# WORLD-SIMULATION AS PRE-TRAINING FOR SCALABLE PERCEPTION

## ABSTRACT

Image-based autoregressive next-token prediction offers a promising avenue for developing world video simulators for autonomous driving. However, applications of these autoregressive models for common perception tasks such as geometric and semantic understanding remains under-explored, largely due to the difficulty of applying discrete token modeling to perception tasks. In this paper, we introduce PerceptionLM, an end-to-end framework that leverages autoregressive world simulators to effectively improve Perception tasks. It consists of a token-based pretraining stage and a novel fine-tuning stage that adapts discrete tokens to continuous embeddings for perception tasks. During pretraining, we leverage the world knowledge from Segment Anything and Depth Anything through autoregressive next-token prediction to imbue the model with world knowledge from multiple vision modalities. During fine-tuning, we propose a novel decoder adaptor to fuse discrete tokens with continuous embeddings from image encoders, which overcomes the limitations of discrete tokens. With PerceptionLM, we observe impressive scaling properties, where quality is consistently improved when providing more training compute or longer temporal context. On multiple public benchmarks including nuScenes, nuImages, Waymo Open Dataset, and Waymo Open Motion Dataset, PerceptionLM demonstrates significant performance improvements for common perception tasks such as depth estimation and semantic segmentation, highlighting its potential for scaling vision-only foundation models for autonomous driving.

## 1 INTRODUCTION

Autoregressive Transformers, which perform next-token prediction based on previously observed sequences, have emerged as a powerful class of generative models for language understanding (Radford et al., 2019; Kaplan et al., 2020; Touvron et al., 2023). Previous studies (Zhao et al., 2024; Team et al., 2023; Radford, 2018) have shown that large language models pretrained on Internet-scale data can encapsulate rich world knowledge, making them highly effective for a variety of downstream tasks, including chat bots (Kalyan, 2023; Radford et al., 2019), image or audio comprehension (Pi et al., 2024; Borsos et al., 2023), and code generation (Ugare et al., 2024).

Building upon these capabilities, autoregressive Transformer models have recently been extended to autonomous driving for world modeling (Hu et al., 2023). Trained on large unannotated driving videos that are tokenized into sequences of discrete tokens, these world models predict future image tokens by autoregressively generating next tokens from prior token sequences. Although these models show great promise in generating realistic driving scenarios, prior research has predominantly focused on visual quality. However, the extent to which such image-based autoregressive models inherently capture world knowledge, and how this knowledge can be effectively leveraged to improve perception tasks, such as geometric and semantic understanding, remains under explored.

A major challenge in adapting autoregressive models for perception tasks lies in their reliance on tokenized representations. While discrete tokens provide a compact and unified format for multiple modalities, the tokenization process is inherently error-prone and often unsuitable for certain perception tasks in autonomous driving. For example, tasks such as monocular depth estimation with LiDAR ground-truth supervision involve handling sparse data, making it difficult to accurately represent such signals using discrete tokens. This sparsity can lead to significant information loss

during the tokenization process, limiting the model's ability to reconstruct accurate depth maps. In contrast, classical perception pipelines are more adept at learning sparse signals through their specialized decoders and masked losses, which preserve the integrity of sparse data throughout training. If we can integrate the strengths of both approaches, we can potentially extend the autoregressive world simulator to perform a wider range of tasks while leveraging its capacity to encode and reason over rich world knowledge.

To address these challenges, we introduce PerceptionLM, *i.e*, Perception as Language Modeling, a novel end-to-end framework that integrates continuous perception signals with an Autoregressive Transformer to enhance perception tasks. PerceptionLM comprises two stages: a token-based pretraining stage and a novel fine-tuning stage that bridges discrete tokens with continuous embeddings. During pretraining, we leverage world knowledge from existing models, such as Segment Anything (Kirillov et al., 2023) and Depth Anything (Yang et al., 2024), through a next-token prediction objective, enabling the model to incorporate rich visual information from multiple modalities. In the fine-tuning stage, we employ a novel decoder adaptor that fuses discrete tokens with continuous embeddings from image encoders, overcoming the limitations of relying solely on discrete tokens for perception tasks. Extensive experiments on public benchmarks, including nuScenes, nuImages (Caesar et al., 2020), Waymo Open Dataset (Sun et al., 2020), and Waymo Open Motion Dataset (Chen et al., 2024) demonstrate that PerceptionLM achieves significant performance improvements for both sparsely-supervised perception tasks, such as depth estimation, and densely-supervised tasks, such as semantic segmentation. Our findings highlight the potential of PerceptionLM to scale vision-only foundation models for autonomous driving and utilize world knowledge effectively for perception applications.

In summary, our contributions are:

- We propose a novel pretraining scheme which uses autoregressive future prediction for long temporal sequences, operating on both tokenized images and outputs from foundation models for depth and segmentation.

- We perform extensive scaling experiments to demonstrate the scaling property for our architecture.

- We propose a novel lightweight adaptor that integrates the autoregressive world simulator, which operates on discrete visual tokens, with a convolutional encoder-decoder network for dense predictions.

- We conduct extensive experiments and ablations on the improvements provided by our architecture for downstream tasks in depth estimation and semantic segmentation across multiple datasets, demonstrating the efficacy of our model and the improvements provided by scaling.

## 2 RELATED WORK

**World Simulation** Leveraging physical world simulation has benefited various computer vision tasks. Early works such as MuZero (Schrittwieser et al., 2020) and DreamerV2 (Hafner et al., 2020) laid the foundation by modeling latent dynamics for reinforcement learning, with DreamerV2 showing notable improvements by switching to discrete latents. VAE-RNN (Ha & Schmidhuber, 2018) further refines the world model paradigm by encoding observations with VAE and modeling temporal dynamics with MDN-RNN (Graves, 2013). Recent work, like Drive-WM (Wang et al., 2024) and DriveWorld (Min et al., 2024), adapt world models to visual forecasting and 4D scene understanding, learning spatial-temporal representations to improve perception and planning. Other advancements such as GAIA-1 (Hu et al., 2023) and Copilot4D (Zhang et al., 2023) apply discrete token-based approaches to model sequences of sensor inputs for more effective future state prediction in 2D and 3D spaces, respectively. In parallel, approaches like DreamTeacher (Li et al., 2023) and Diffusion World Model (Ding et al., 2024) demonstrate how diffusion generative models could be used for learning world models. These works underscore the growing importance of world models in enabling scalable and robust world knowledge understanding for general perception tasks across diverse visual domains.

**Perception Task Fine-tuning** It is usually helpful to fine-tune pretrained models on task-specific datasets in order to improve performance of perception tasks. Traditional models like Mask R-

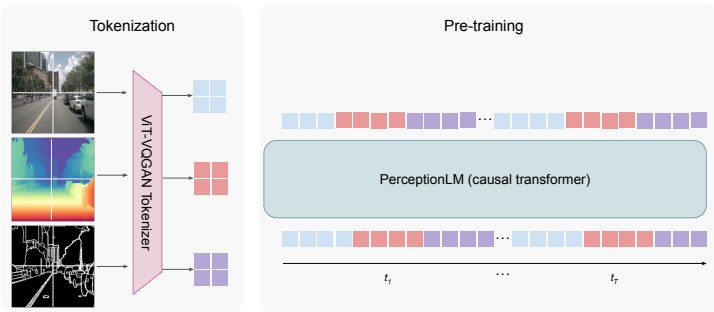

Figure 1: **PerceptionLM pretraining pipeline.** Different modalities (image, depth, and edge) are tokenized though a ViT-VQGAN tokenizer and flattened to one dimension. Tokens from multiple consecutive time steps ($t_1$, ..., $t_T$) are concatenated and used to pretrain the PerceptionLM backbone through autoregressive next-token prediction. Best viewed in color.

CNN (He et al., 2017) and DETR (Carion et al., 2020) fine-tune pretrained vision backbones from large datasets like ImageNet (Deng et al., 2009). LVM (Bai et al., 2024) pretrains a visual tokenizer to generate discrete tokens, then fine-tunes the entire Autoregressive Transformer model for tasks such as segmentation and pose estimation, leveraging the sequential nature of the pretraining. MultiMAE (Bachmann et al., 2022), on the other hand, conducts fine-tuning with task-specific decoders for tasks like depth estimation and segmentation. In RoboLLM (Long et al., 2024), a pretrained plain-backbone from a multi-modal language model is fine-tuned for robotic vision tasks, also with task-specific heads introduced during fine-tuning to adapt the general pretrained backbone to specific tasks like object segmentation. VLMwRL (Zhai et al., 2024) combines reinforcement learning with pretrained vision-language models, adding Chain-of-Thought (Wei et al., 2022) reasoning during fine-tuning to enhance decision-making in goal-directed tasks. Each of these methods shows how pretrained models can be adapted for downstream perception tasks through different strategies, such as task-specific heads, reinforcement learning, or direct fine-tuning of the backbone. Our method PerceptionLM differs by combining the discrete token-based world model with continuous embeddings during fine-tuning, making it versatile for perception tasks requiring both geometric and semantic understanding capabilities.

**Autoregressive Visual Modeling** Initially applied in language models like GPT (Vaswani et al., 2023; Radford et al., 2019; Team et al., 2023) for next-token prediction, Autoregressive (AR) modeling has extended to vision tasks, tokenizing images and videos into sequences for generation (Yu et al., 2022; Sun et al., 2024; Ge et al., 2022). The AR universal learner paper (Malach, 2023) further demonstrates that even simple autoregressive models can approximate complex functions like those computable by Turing machines, highlighting the power of next-token prediction beyond just architecture. To tackle scalability in high-resolution image generation tasks, VAR (Tian et al., 2024) introduces multi-scale token prediction, improving efficiency and quality by predicting at different image resolutions. Expanding into multi-modality, 4M and 4M-21 (Mizrahi et al., 2024; Bachmann et al., 2024) extend order-agnostic masked AR models to handle a variety of tasks across multiple modalities, enabling better generalization across tasks like text, image, and depth prediction through masked modeling strategies. On the temporal front, VideoPoet (Kondratyuk et al., 2023) applies next-token prediction for zero-shot video generation, generating coherent video sequences by predicting video tokens over time. Transfusion (Zhou et al., 2024) combines the diffusion loss with autoregressive transformer to train on mixed-modality data. Inspired by the success of tokenized pre-training in the large language domain, where discrete representations have proven effective in capturing structured world knowledge, we aim to further explore the potential of discrete autoregressive pretraining for vision tasks with mixed discrete and continuous tokens.

# 3 METHOD

In this section, we detail our method, which consists of the PerceptionLM architecture, as well as the pipeline to integrate PerceptionLM with a classical perception encoder-decoder model.

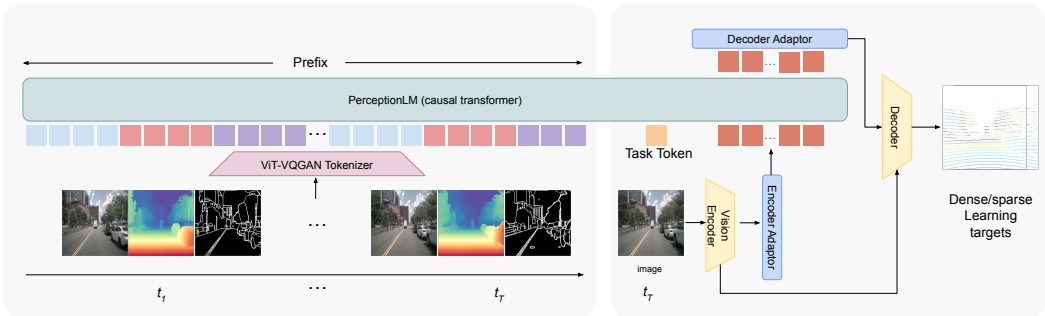

Figure 2: **PerceptionLM fine-tuning pipeline.** In addition to the pretraining setup, during fine tuning, an ImageNet-pretrained ConvNeXt model is adopted as `vision encoder` to extract task specific multi-scale visual features. These features are aligned with multi-frame VQ tokens through an `encoder adapter`. The flattened visual features are fed into the PerceptionLM causal Transformer along with a task specific token. The extracted features are passed to a `decoder adapter` followed by a task-specific `decoder` head to produce the task outputs. Best viewed in color.

The pipeline includes pretraining and fine-tuning stages. During pretraining (Fig. 1), we first construct visual tokens for driving videos with 1 billion images, and then use these tokens to pretrain PerceptionLM with the next token prediction task. During fine-tuning (Fig. 2), besides the visual tokens used in the pretraining stage, we also add task tokens and visual queries from a convolutional encoder to perform perception tasks. In the rest of the section, we will provide more details about the visual token construction (Sec. 3.1), model architecture (Sec. 3.2), and training regime (Sec. 3.3).

### 3.1 GENERATING A LARGE SCALE PRETRAINING DATASET WITH WORLD KNOWLEDGE

A high quality dataset is key to any large scale model training. In this work, we curate a large scale private dataset of driving videos with 1 billion images, on top of which we generate additional modalities from existing open-source foundation models, which we tokenize and combine with the original images.

**Imparting World Knowledge:** While it has been shown that these models that predict future images can learn to reason about the world beyond visual appearance, we posit that learning can be further improved by adding other modalities such as geometry and semantics explicitly into the model training. In order to augment the reasoning into other modalities, we leverage existing foundation models to generate high-quality pseudo-labels, facilitating the creation of large scale, aligned multi-modal data. In particular, we posit that a strong, scalable perception model needs to be able to process both geometric and semantic information about a scene. In this work, we propose to directly imbue the model with this information by adding additional modalities which represent geometry and semantics to the set of tokens our model reasons on. In particular, we adopt Depth Anything (Yang et al., 2024) to generate relative depth maps in order to encode a geometric view of the scene, and Segment Anything to produce segmentation masks, to encode a semantic view of the scene. As the order of the masks is permutation invariant, we opt to instead produce edge maps (*i.e.* a binary mask for edge region) from the segmentation masks in order to retain a consistent set of labels. While other text-aligned foundation models like CLIP (Radford et al., 2021) are available, we opt to use dense per-pixel models in order to retain a consistent output representation for tokenization. In order to enable scaling of our model to billions of parameters, we curate an internal dataset with one billion driving images, on which we generate the corresponding relative depth and edge maps.

**Tokenization:** To unify the representation space for multiple modalities, we adopt a pretrained ViT-VQGAN (Esser et al., 2021; Yu et al., 2021) to convert each modality (image, depth, edge) into a sequence of discrete tokens. For the depth and edge modalities, we broadcast along the channel dimension to form a RGB image, and normalize the values to [0, 1] before tokenization. We carefully select modalities for pretraining and fine-tuning to ensure that the fine-tuning task is either unseen (e.g., semantic segmentation vs edge detection) or represented differently (e.g., absolute depth vs relative depth). The tokenization process is pre-computed for reuse, resulting in a substantial corpus of multi-modal visual token sequences.

## 3.2 PERCEPTIONLM ARCHITECTURE

As shown in Fig. 2, PerceptionLM consists of a *causal Transformer backbone* that reasons about the interactions between multi-modal temporal context and continuous visual signals, a canonical *encoder-decoder* architecture for extracting continuous visual signals and performing a wide range of perception tasks, and *adapters* that effectively align the discrete visual tokens and continuous visual features.

**Causal Transformer backbone:** The causal Transformer processes three key inputs.

The first is a set of discrete visual tokens, which encodes multi-modal temporal cues. These tokens are generated by the concatenation of the image, depth, and edge map tokens over multiple temporal frames of our curated dataset. During pretraining, they are used as a supervision signal, where we train the model on the next-token prediction task to efficiently encode the multi-modal temporal information. In the fine-tuning phase, these tokens act as a prefix to provide additional multi-modal and temporal context for historical frames.

The second and third inputs are learnable queries, providing the model with task-specific information and visual cues. In this work, we use the intermediate features from a convolutional encoder as the visual queries. As these features have a spatial correspondence, they can be thought of as input-dependent position embeddings for each spatial location. These inputs are only provided during fine-tuning, when the model is expected to produce outputs specific to each downstream task.

To combine the discrete and continuous tokens for the Transformer, all discrete tokens from the first input are mapped to continuous embeddings via a learned lookup table. These embeddings are then concatenated with the continuous task and visual query tokens, forming a unified sequence that is input to the Transformer:

$$\{\hat{q}_i\}_{i=1}^L = \texttt{Transformer}\left(\left[\phi\left(\underbrace{\{\{I_t^i\}_{i=1}^L, \{D_t^i\}_{i=1}^L, \{E_t^i\}_{i=1}^L\}_{t=1}^T}_{\text{prefix}}\right), \underbrace{Q_{task}}_{\text{task token}}, \underbrace{\{q_i\}_{i=1}^L}_{\text{queries}}\right]\right) \quad (1)$$

where $\{\{I_t^i\}_{i=1}^L, \{D_t^i\}_{i=1}^L, \{E_t^i\}_{i=1}^L\}_{t=1}^T$ are the visual tokens. $\{I_t^i\}_{i=1}^L$ is the set of tokens for image modality with length $L$, $\{D_t^i\}_{i=1}^L$ is the set of depth tokens, $\{E_t^i\}_{i=1}^L$ is the set of edge tokens and $T$ is the temporal length. $\{q_i\}_{i=1}^L$ is the set of visual query embeddings and $\{\hat{q}_i\}_{i=1}^L$ is the updated query features from the last layer of the Transformer block. $\phi$ represents the lookup table that maps discrete token to continuous embeddings and $Q_{task}$ is the task token that instructs the model to perform tasks.

In this work, we focus on fine-tuning for a single task; however the framework is generalizable and can be extended to instruct the model to perform multiple tasks simultaneously (Kondratyuk et al., 2023). This extension is left for future work.

**Vision Encoder-Decoder:** For downstream tasks, we adopt a classical convolutional encoder-decoder architecture. For the encoder, we use an off-the-shelf ConvNeXt (Liu et al., 2022) to extract multi-scale visual features $\{F_i\}_{i=1}^N$ where $F_i \in \mathcal{R}^{H_i \times W_i \times C_i}$. These features are then fed into a convolutional decoder to produce dense per-pixel outputs. In this work, the convolutional decoder follows the architecture in Li et al. (2021) where multi-resolution feature maps are gradually up-sampled and interacted with each other via concatenation and convolutions to produce the final estimation.

**Encoder Adapter:** In order to augment this encoder-decoder with PerceptionLM, we propose adapters for the encoder and decoder to fuse the two sets of embeddings. In the encoder, the discrete token embeddings and the extracted visual features reside in distinct feature spaces. To align these spaces, we introduce an adapter that projects the continuous visual embeddings into the discrete token embedding feature space. More specifically, the multi-scale feature maps are resized using bilinear interpolation to ensure a consistent feature size. These resized feature maps are then concatenated along the channel dimension and passed through a few convolutional layers in EncoderAdaptor for feature refinement:

$$Q = \texttt{EncoderAdaptor}\left(\texttt{Concat}\left(\{\texttt{Bilinear}(F_i)\}_{i=1}^N\right)\right) \in \mathcal{R}^{H \times W \times C} \quad (2)$$

where $F_i$ is the multi-scale features from encoder, $N$ is the number of feature scales and $HW$ is equal to the sequence length $L$ to match the spatial dimension in pretraining. The refined feature map Q is subsequently flattened along the spatial dimension to form a sequence of continuous embeddings $\{q_i\}_{i=1}^L$ in equation 1 and acts as the visual queries for the Transformer.

**Decoder adapter:** The updated query features $\{\hat{q}_i\}_{i=1}^L$ in equation 1 are reshaped back to a feature map $\hat{Q} \in \mathcal{R}^{H \times W \times C}$. In order to imbue the world knowledge from the causal Transformer, we concatenate the output feature $\hat{Q}$ with the original multi-scale feature $F_i$ from vision encoder. The imbued features are then passed through a few convolutional layers in DecoderAdaptor to produce the adapted feature $\hat{F}_i$ for task-specific decoders.

$$\hat{F}_i = \texttt{DecoderAdaptor}\left(\texttt{Concat}\left(F_i, \texttt{Bilinear}\left(\hat{Q}\right)\right)\right), \forall i = 1, \dots, N \qquad (3)$$

### 3.3 Training Regime

**Pretraining:** Pretraining is essential to imbue the model with foundational knowledge for reasoning generalizable feature representation, enabling it to capture complex visual patterns and relationships, before fine-tuning on task-specific data. To this end, we pretrain the large vision model on our internal dataset of large scale driving videos with image tokens and inferred depth and edge tokens. As discussed in Sec. 3.1, all modalities are tokenized to form the visual tokens. The autoregressive Transformer is then trained on the next-token prediction task, allowing the model to effectively learn and reason about the interactions among these discrete tokens, utimately encapsulating rich world knowledge within. In particular, we train the model to predict the discrete token ID of next token via N-way classification and a cross-entropy loss.

**Fine-tuning:** After pretraining, we fine-tune the entire PerceptionLM in an end-to-end manner, instructing the Transformer to perform downstream perception tasks that are usually challenging for token-based architectures. Task-specific heads and corresponding loss functions are employed to guide the learning process. For the depth estimation, LiDAR point clouds are projected onto images as supervision signals, and an L1 loss is applied. For semantic segmentation, a sigmoid focal cross-entropy loss is employed.

## 4 Experiments

### 4.1 Datasets

We evaluate our method on several autonomous driving datasets to verify the efficacy of both our pretraining pipeline and its effects on downstream perception tasks in depth estimation and semantic segmentation. For large scale pretraining, we curate an internal dataset for autoregressive next-token prediction task learning and then fine tune and evaluate our method on open datasets for depth and semantic segmentation tasks.

We evaluate our method on depth estimation on the nuScenes, WOMD, and WOD datasets. To do so, we take the LiDAR scans corresponding to each image, and project each lidar point into the image to establish a ground-truth depth for the corresponding pixel. We then train our model to predict per-pixel depth where only sparse positions with ground-truth depth values will be used to supervise the learning and evaluation. For semantic segmentation, we leverage the semantic segmentation groundtruth in the nuImages dataset to directly supervise a multi-class classification model.

**Pretraining dataset:** To evaluate PerceptionLM autoregressive pretraining at scale, we curate an internal dataset with one billion driving images at 5Hz without labels.

**nuScenes:** nuScenes is a public large scale 3D dataset for autonomous driving with 12 HZ images and 2HZ annotation frequency. The dataset consists of 1,000 driving sequences, with 1.4 million camera images and 390k LiDAR sweeps.

**nuImages:** nuImages is a public 2D autonomous driving dataset for image based detection and semantic segmentation. The dataset contains 93k video clips with 13 frames each, spaced out at

2Hz. Each video contains a single image labeled with segmentation masks, with 100k semantic segmentation masks in total.

**Waymo Open Dataset (WOD)** The Waymo Open Dataset is a large scale autonomous driving dataset with 1150 driving sequences with 5 camera images and paired LiDAR sweeps at each frame. Each sequence consists of 20s of data recorded at 10Hz.

**Waymo Open Motion Dataset (WOMD)** In addition to the Waymo Open Dataset, we also evaluate on the Waymo Open Motion Dataset. WOMD is a much larger dataset, consisting of over 100,000 scenes, each 20 seconds long at 10Hz. The dataset contains both camera sensor tokens and LiDAR sweeps for each frame We asked Waymo for permission to access the raw images from WOMD and regenerated the tokens for the fine-tuning tasks.

In this work, we resize all images to 256×256 and apply temporal subsampling, resulting in 5 HZ for pretraining data, 6HZ for nuScenes, 2HZ for nuImages, 5HZ for WOD, and 5HZ for WOMD.

## 4.2 Experimental setup

**Pretraining:** We conduct experiments with 1, 2, 4, and 8 frames of temporal context window, where each frame contains 3 modalities: image, depth and edge, and each modality is tokenized into 1024 tokens. The total number of input tokens are 3072, 6144, 12288, and 24576 for the 1, 2, 4, and 8-frame experiments, respectively, We use a decoder only Transformer with causal masking, RoPe positional embedding and RMS normalization (Vaswani, 2017; Su et al., 2024; Zhang & Sennrich, 2019). To efficiently model long context lengths, we use a Blockwise Parallel Transformer with block size 1024 (Liu & Abbeel, 2024) for our casual Transformer backbone, which reduces the memory and time complexity from quadratic to linear with respect to input sequence length. For scaling 1 and 2-frame models, we jointly scale hidden layer size=128×scale, intermediate size = 256×scale, and number of hidden layers = 4×scale, where scale $\in [1, 2, 3, 4, 6, 8, 10, 12]$, leading to models with 2.7M, 9.5M, 24.1M, 50.5M, 154M, 352M, 677M, and 1.1B parameters. For 4-frame and 8-frame models, we scale model up to scale = 10 and 8 respectively.

**Fine-tuning:** ConvNeXt-S (Liu et al., 2022) is adopted as the vision backbone for both depth prediction and semantic segmentation. ConvNeXt-S is very light-weight (50M params) and can be easily integrated with PerceptionLM with negligible computation overhead, as compared with our largest 1.1B model. In this work, we initialize our ConvNeXt model with the weights pretrained on imageNet-22K and fine-tuned on ImageNet1-K (Deng et al., 2009). By default, we adopt three modalities (image, relative depth, edge map) and eight frames to form the visual tokens. For consistency with pretraining, the input image to ConvNeXt is resized to a shape of (256, 256, 3), and the induced intermediate visual queries share the shape of (32, 32, $C$) where the $C$ matches with the Transformer's hidden layer size. For experiments with PerceptionLM, we also initialize the visual encoder and decoder weights with the encoder-decoder only model trained on the fine-tuning task. All models are trained on 64 TPUv5e with per-device batch size 1. For ablation studies, we evaluate the depth prediction on NuScenes dataset and semantic segmentation on nuImages dataset.

## 4.3 Scaling up pretraining.

We first systematically study the scaling properties of our pre-training in both model capacity and temporal context length.

**Beyond Model Parameters – Scaling Temporal Context:** While model parameters are important, scaling perception reasoning also critically depends on the length of the temporal context. Increasing the number of temporal frames allows PerceptionLM to incorporate more contextual information from the history. However, this increased reasoning capacity is not reflected in the sheer number of parameters. To accurately capture PerceptionLM's true capacity, we use model training FLOPs (floating-point operations) as the metric. This measures the computational effort required to train the model, which better reflects the ability to process longer temporal contexts.

**Scaling Model Capacity:** Table 1 shows the loss trend fitting. We fit the next token prediction cross-entropy loss with respect to training GigaFLOPs to a power law, i.e., $L = A/GFLOPs^B + C$, where A and B captures the scaling trends and C indicates the irreducible loss. Similar to Large

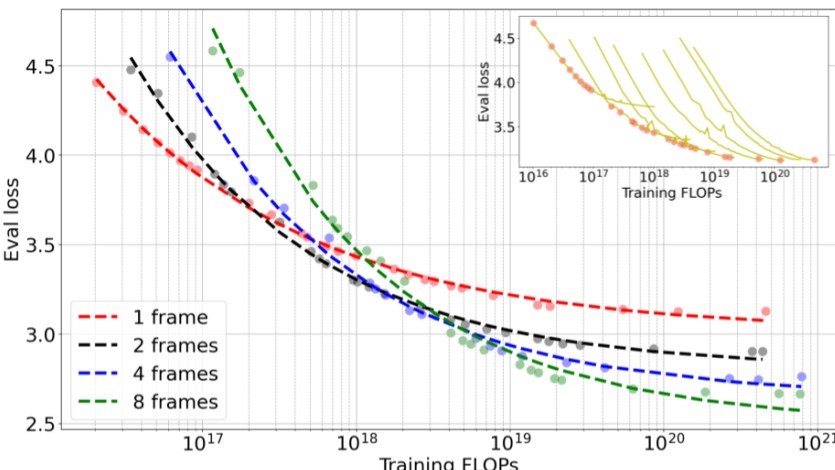

Figure 3: **Scaling trends for PerceptionLM pretraining.** Evaluation loss is defined as the cross-entropy loss of next-token prediction, shown in Figure 1. We fit the evaluation losses (dotted points) to power law functions and display the corresponding fitted dashed line, as in Table 1. The right top inserted figure shows the actual training loss curves for the 1-frame model, where only those red dotted points on the lowest envelope are kept for the main figure. Best viewed in color.

Table 1: **Temporal and model scaling fitting.** Evaluation loss is fitted to a power law function $L = A/GFLOPs^B + C$ with respect to the training GigaFLOPs. In general, longer temporal context tends to have lower irreducible eval loss.

| Num. of frames | 1 | 2 | 4 | 8 |
|---|---|---|---|---|
| Scaling curve | $L_1 = \dfrac{0.419}{\text{GFLOPs}^{0.312}} + 3.01$ | $L_2 = \dfrac{0.498}{\text{GFLOPs}^{0.370}} + 2.81$ | $L_3 = \dfrac{0.677}{\text{GFLOPs}^{0.375}} + 2.65$ | $L_4 = \dfrac{0.970}{\text{GFLOPs}^{0.383}} + 2.50$ |

Language Models, scaling the model capacity with more training FLOPs leads to lower next token prediction loss.

**Scaling Temporal Context:** As indicate from both Figure 3 and Table 1, pretraining a PerceptionLM with high FLOPs only on a single frame leads to a higher irreducible loss (fitting parameters C) 3.01 compared to pretraining with 2, 4, and 8 frames. This highlights the need for incorporating longer visual context information to improve next token prediction.

**Scaling PerceptionLM: Balancing Model and Temporal Context** While increasing either model capacity or temporal context can improve performance, our analysis reveals that a balanced approach is crucial for optimal results. Surprisingly, simply expanding the context window with a small model actually increases the evaluation loss. This is evident in the loss scaling factor (A), which grows from 0.419 for the single-frame model to 0.498, 0.677, and 0.970 for the 2, 4 and 8-frame models, respectively.

This behavior stems from the unique structure of PerceptionLM. Unlike traditional language models that process sequential tokens, PerceptionLM handles tokens from a four-dimensional space (time, modality, row, and column). These tokens exhibit localized correlations within this multi-dimensional space, with a token at position $(t, m, r, c)$ being more closely related to its immediate neighbors $(t \pm 1, m \pm 1, r \pm 1, c \pm 1)$. While flattening these dimensions into a single sequence simplifies next-token prediction, it disrupts these inherent relationships. Consequently, it's essential to scale both the model capacity and the temporal context length concurrently. This is further supported by the GFLOPs exponent (parameter B), where the 8-frame model exhibits a larger decaying factor (0.383) compared to the single-frame model (0.312), indicating that scaling the model leads to faster loss reduction with longer context lengths.

**Compute optimal scaling:** Based on fittings in Figure 3 and Table 1, we find 1, 2, 4, and 8 frames model evaluation loss curves intersect exactly one time with each other, indicating the shifting of optimal training paradigm, i.e. for a small training budget $< 10^{17}$ FLOPs, single frame model is more compute optimal, while for a large training budget $> 10^{19}$ FLOPs, 8-frame model is more efficient.

Table 2: **Benchmark results for depth prediction on nuScenes, WOD and WOMD datasets** with 352M PerceptionLM and 8 frames.

| Methods | Datasets | AbsRel↓ | RMSE↓ | RMSE log↓ | $\delta_1$↑ | $\delta_2$↑ | $\delta_3$↑ |
|---|---|---|---|---|---|---|---|
| ConvNeXt | nuScenes | 0.067 | 3.305 | 0.148 | 0.968 | 0.979 | 0.985 |
| PerceptionLM w/o pretrain | nuScenes | 0.067 (0.0%) | 3.382 (+2.3%) | 0.146 (-1.4%) | 0.97 | 0.981 | 0.986 |
| PerceptionLM | nuScenes | **0.054 (-19.4%)** | **2.951 (-10.7%)** | **0.125 (-15.5%)** | **0.978** | **0.986** | **0.990** |
| ConvNeXt | WOMD | 0.069 | 4.461 | 0.137 | 0.936 | 0.980 | 0.992 |
| PerceptionLM w/o pretrain | WOMD | 0.063 (-9.4%) | 4.267 (-4.3%) | 0.131 (-0.9%) | 0.945 | 0.982 | 0.992 |
| PerceptionLM | WOMD | **0.057 (-17.4%)** | **3.940 (-11.7%)** | **0.121 (-11.7%)** | **0.955** | **0.985** | **0.993** |
| ConvNeXt | WOD | 0.102 | 5.318 | 0.171 | 0.882 | 0.967 | 0.989 |
| PerceptionLM w/o pretrain | WOD | 0.098 (-3.9%) | 5.223 (-1.8%) | 0.166 (-2.9%) | 0.890 | 0.970 | 0.989 |
| PerceptionLM | WOD | **0.089 (-12.7%)** | **4.948 (-7.0%)** | **0.156 (-8.8%)** | **0.908** | **0.974** | **0.990** |

Table 3: **Semantic Segmentation results on nuImage dataset.**

| Methods | mIoU↑ |
|---|---|
| ConvNeXt | 65.21 |
| PerceptionLM w/o pretraining | 65.06 |
| PerceptionLM | **67.35** |

Table 4: **Fine-tuning inputs study** based on a PerceptionLM with 352M parameters and 8 frames.

| Fine-tuning inputs | AbsRel↓ | RMSE↓ | $\delta_1$↑ |
|---|---|---|---|
| Discrete tokens only | 0.062 | 3.159 | 0.972 |
| Discrete tokens + continuous embeddings | **0.054** | **2.951** | **0.978** |

This finding suggests that model capacity and temporal context should be scaled harmoniously to maximize performance gains and minimize the training compute.

## 4.4 FINE-TUNING EXPERIMENTS

**Main performance comparisons:** Table 2 reports the benchmark results for depth prediction on the nuScenes, WOD, and WOMD datasets, while Table 3 presents the semantic segmentation benchmarks on nuImages. PerceptionLM demonstrates a substantial improvement over the baseline ConvNext model, suggesting that the proposed framework effectively extracts rich world knowledge to enhance perception performance. In contrast, PerceptionLM without pretraining exhibits comparable or slightly lower performance relative to the baselines, which we attribute to the limited size of the available datasets. The discrete token context is represented as a sequence of integer values, resulting in a highly compact representation. This makes pretraining on large-scale datasets crucial, as the model otherwise struggles to extract meaningful information from these token streams due to the constrained amount of training data.

**Fine tuning architecture ablation:** A naive alternative fine-tuning approach is to simply use the same pre-training method that tokenize all inputs into discrete tokens, without adding the extra ConvNeXt vision encoder with continuous embeddings in our PerceptionLM. In particular, one can add a list of randomly initialized queries and rely on the discrete tokens from Prefix in Figure 2 to produce all the necessary features for the decoder.

Table 4 compares these two approaches, where the baseline relies solely on discrete tokens but our PerceptionLM relies on both discrete tokens from Prefix and continous embeddings from a ConvNeXt vision encoder. Results show our PerceptionLM achieves much better quality than the baseline, highlighting the importance of combining both scene dependent embeddings and discrete tokens for Perception tasks.

**Model scaling experiment:** Table 5 presents the results of the scaling experiment during fine-tuning. Increasing the model size generally leads to improved performance, demonstrating the strong scalability of PerceptionLM.

**Temporal context length:** Table 6 examines the quality impact of temporal context length. Results show that increasing the sequence length consistently improves model performance, indicating that PerceptionLM effectively captures temporal cues to support downstream perception tasks.

**Scaling ConvNeXt comparison:** In Table 7, we investigate the performance gains from scaling a specialized ConvNeXt model vs our proposed method. As shown in the table, ConvNeXt performance peaks at around 203M parameters, without further gains when scaling to 357M parameters.

Table 5: **Model scaling** with single frame.

| Scale | AbsRel↓ | RMSE↓ | $\delta_1$ ↑ | mIoU↑ |
|---|---|---|---|---|
| 50M | 0.063 | 3.178 | 0.971 | 66.38 |
| 352M | 0.060 | 3.047 | 0.973 | 66.79 |
| 1.1B | **0.059** | **2.997** | **0.973** | **67.08** |

Table 6: **Influence of temporal context.**

| #Frames | AbsRel↓ | RMSE↓ | $\delta_1$ ↑ | mIoU↑ |
|---|---|---|---|---|
| 1 | 0.060 | 3.047 | 0.973 | 66.79 |
| 4 | 0.057 | 2.966 | 0.976 | 67.28 |
| 8 | **0.054** | **2.951** | **0.978** | **67.35** |

Table 7: **ConvNeXt encoder scaling**.

| Model | # Params | RMSE↓ |
|---|---|---|
| ConvNeXt-T | 32M | 3.541 |
| ConvNeXt-S | 53.6M | 3.305 |
| ConvNeXt-L | 203M | 3.167 |
| ConvNeXt-XL | 357M | 3.212 |
| ConvNeXt-S + PerceptionLM | 448M | **2.951** |

Table 8: **Inputs modality ablations**.

| Image | Edge | Depth | AbsRel↓ | RMSE↓ | $\delta_1$ ↑ |
|---|---|---|---|---|---|
| | | | 0.067 | 3.305 | 0.968 |
| ✓ | | | 0.064 | 3.204 | 0.971 |
| | ✓ | | 0.063 | 3.183 | 0.971 |
| | | ✓ | 0.060 | 3.076 | 0.972 |
| ✓ | ✓ | ✓ | **0.057** | **2.966** | **0.976** |

Table 9: **Fine-tuning strategy ablations**. 🔥 denotes trainable; ❄ denotes frozen. Fine-tuning all weights leads to the best quality.

| Encoder | Deoder | PerceptionLM | AbsRel↓ | RMSE↓ | $\delta_1$ ↑ |
|---|---|---|---|---|---|
| ❄ | ❄ | ❄ | 0.065 | 3.239 | 0.970 |
| ❄ | 🔥 | ❄ | 0.065 | 3.243 | 0.971 |
| 🔥 | 🔥 | ❄ | 0.063 | 3.192 | 0.971 |
| 🔥 | 🔥 | 🔥 | **0.057** | **2.966** | **0.976** |

In contrast, our proposed method shows continued improvement in scaling as shown in Table 5, further demonstrating its scalability.

**Modality ablations:** In classical computer vision, modalities that are more closely aligned with the target task tend to contribute more significantly to the final performance. In PerceptionLM, all multi-modal information is represented as discrete tokens, i.e., discrete integers. Table 8 investigates whether this synergy persists for discrete modality inputs. The results indicate that relative depth tokens contribute the most to absolute depth prediction, reflecting a similar relationship as observed in classical vision tasks. Additionally, the model utilizing edge modality inputs outperforms the one relying on raw image tokens. The best performance is achieved when incorporating all three modalities, highlighting the effectiveness of multi-modal fusion in PerceptionLM.

**Fine-tuning strategy:** To identify the most effective training strategy for optimizing the alignment between discrete token embeddings and continuous visual signals, we evaluate several configurations: (1) frozen encoder, decoder, and Transformer with only the adapters being learnable, (2) frozen encoder and Transformer with trainable adapters and decoder, (3) trainable encoder and decoder with a frozen Transformer, and (4) a fully trainable model. As shown in Table 9, the fully trainable configuration of PerceptionLM achieves the highest performance, underscoring the importance of jointly training all model components to achieve optimal results.

## 4.5 CONCLUSION

In this work, we investigate how to leverage world simulation to enhance perception tasks that are traditionally hard to be represented as discrete tokens. As a solution, we propose PerceptionLM, an end-to-end framework that integrates continuous perception signals with an auto-regressive Transformer to enhance perception tasks. Extensive scaling experiments are conducted to understand the scaling properties of PerceptionLM in terms of both model capacity and multi-modal temporal context length. Comprehensive experiments on public datasets demonstrate significant improvements when PerceptionLM is integrated.

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
