# OpenReview forum: "World-simulation as pre-training for scalable perception"
_ICLR.cc/2025/Conference — Submitted to ICLR 2025_

### Official Review · Reviewer_1iJo · 2024-10-30

**Soundness:** 2
**Presentation:** 2
**Contribution:** 2
**Rating:** 5
**Confidence:** 4

**Summary:**

This paper addresses the issue of information loss in discrete tokens by proposing a method to incorporate continuous features into discrete features, and experiments demonstrate the effectiveness of the approach.

**Strengths:**

The ablation studies demonstrate the effectiveness of the author's method.

The writing is good.

**Weaknesses:**

1.	I have some questions about the motivation of the paper. I don't believe that discrete tokens lead to significant information loss. The image is already a form of discrete representation, and this discrete representation is used in almost all computer vision tasks. If discretized tokens suffer significant information loss, it may be caused by the inappropriate length or other settings of the codebook. Can you provide more concrete evidence or examples of where discrete token representations specifically lead to information loss in your approach?

2.	There is a lack of comparison with other world model methods. Predicting the next token is a common approach in the area of world model, and many papers have addressed related topics, such as Drive-WM[1] and Vista[2]. The authors should explain how their method differs from or improves upon specific aspects of Drive-WM and Vista and the comparison with these methods is needed to demonstrate the effectiveness of the proposed method.

3.	The experiments are primarily ablation studies, lacking comparisons with the latest methods of downstream tasks, making it unclear whether the authors' approach is state-of-the-art. A few key recent methods in depth estimation and semantic segmentation should be included as baselines.

4.	The visualization results of depth estimation and semantic segmentation (versus baselines or other sota methods) can be added to help readers understand more clearly about the advantages of the proposed method.

[1] Driving into the Future: Multiview Visual Forecasting and Planning with World Model for Autonomous Driving.

[2] Vista: A Generalizable Driving World Model with High Fidelity and Versatile Controllability.

**Questions:**

see weaknesses above

---

### Official Review · Reviewer_2Sth · 2024-11-04

**Soundness:** 2
**Presentation:** 3
**Contribution:** 2
**Rating:** 5
**Confidence:** 4

**Summary:**

This work presents PerceptionLM, an approach that uses world simulation (next token prediction) for pretraining representations for perception tasks.
The method employs a two-stage approach: first using ViT-VQGAN tokenization for image, depth, and edge features, followed by autoregressive future prediction using a Blockwise Parallel Transformer (BPT) as the causal model.
They evaluate their pretrained representations on depth estimation and semantic segmentation tasks using a ConvNeXt-S backbone.
The model is trained on a internal billion-image dataset and evaluated across multiple autonomous driving datasets: nuScenes, nuImages, Waymo Open and the Waymo Open Motion Dataset.

**Strengths:**

Comprehensive evaluation across multiple autonomous driving datasets (nuScenes, nuImages, Waymo Open, Waymo Open Motion) demonstrates broad applicability.

Thorough ablation studies examine both model scaling effects and the impact of different input modalities on depth prediction performance, providing valuable insights into the method's behavior.

The scale of pretraining is impressive, utilizing a billion-image dataset which potentially enables better generalization and robustness.

Writing is sound and well-structured.
The paper is easy to follow and the technical content is clear.

**Weaknesses:**

The evaluation tasks (semantic segmentation and absolute depth prediction) are very similar to the pretraining tasks (edge detection and relative depth estimation).
This makes it difficult to assess if the learned representations genuinely generalize beyond the pretraining objectives.

Tables 2 and 3 only compare against ConvNeXt (the vision backbone) as a baseline.
Comparisons against specialist models or other large-scale pretrained autoregressive approaches would better contextualize the method's effectiveness.

There is a notable disconnect between the paper's motivation and evaluation.
While the introduction frames the work as advancing general perception capabilities, the evaluation is focused on segmentation and primarily depth estimation tasks in autonomous driving datasets.

**Questions:**

In Table 2, PerceptionLM uses 8 frames for prediction - how many frames does the baseline ConvNeXt use?

What motivated the choice of edge detection and relative depth as auxiliary tasks during pretraining?
Were other tasks considered that might enable learning more general visual representations?

Some dataset details are missing - specifically the number of videos/sequences in the pretraining set and whether this data will be publicly available.

---

### Official Review · Reviewer_iDsq · 2024-11-04

**Soundness:** 3
**Presentation:** 3
**Contribution:** 2
**Rating:** 5
**Confidence:** 3

**Summary:**

The paper introduces PerceptionLM, an end-to-end framework that leverages autoregressive transformers to enhance perception tasks in autonomous driving. The main contribution of this paper is to demonstrate that pretraining using next-token prediction for world observation on large-scale driving datasets can benefit depth estimation and semantic segmentation tasks. The authors propose a token-based pretraining approach with a novel decoder adaptor for adapting discrete tokens to continuous embeddings during fine-tuning. This approach shows promising results on public benchmarks like nuScenes, nuImages, Waymo Open Dataset, and Waymo Open Motion Dataset, emphasizing scalability in terms of training compute and longer temporal context.

**Strengths:**

- This paper is well-written and easy to follow.
- The proposed framework is well-motivated and coherent.
- The results look good compared to the baseline. The ablation studies are comprehensive.

**Weaknesses:**

- The comparison to existing methods feels incomplete. While ConvNeXt is used as a baseline, there is a lack of broader context that includes other competitive models for perception tasks, especially those utilizing attention mechanisms, self-supervised pretraining, and additional modalities.
- While the authors provide a comparison with ConvNeXt for different scales, it is worth noting that the ConvNeXt baseline only utilizes a single frame and image modality, whereas the proposed method uses additional edge and depth inputs with up to eight frames. This disparity makes the comparison unfair and complicates the readers' ability to understand the real performance gain within the same computation and parameter budget for the proposed method. It would be beneficial if the authors could also discuss the training cost.

**Questions:**

- The paper focuses heavily on autonomous driving datasets. It would be advantageous for the authors to explore other domains, such as pretraining on general videos and fine-tuning on task-specific datasets, to provide more confidence in the model's general applicability.
- For concerns regarding the baseline and runtime/training costs, please refer to the weaknesses section.

I appreciate the authors' efforts in developing such an autoregressive pretraining model. However, the current results lack detailed comparisons with more common baselines, which prevents me from giving a higher rating.

---

### Official Review · Reviewer_NmvC · 2024-11-09

**Soundness:** 3
**Presentation:** 3
**Contribution:** 3
**Rating:** 5
**Confidence:** 3

**Summary:**

This paper introduces PerceptionLM, an end-to-end framework that uses pre-trained world models to enhance perception tasks. During pretraining, the model leverages videos and foundation model knowledge to learn a next-token prediction task on multi-modality inputs. In the fine-tuning stage, a decoder adaptor is learned to integrate discrete tokens with continuous embeddings from image encoders for dense prediction tasks. The authors perform extensive experiments to examine the scaling property of PerceptionLM in terms of model capacity and temporal context length. The authors also conduct fine-tuning experiments for downstream tasks such as LiDAR depth prediction and image semantic segmentation.

**Strengths:**

* This paper studies an interesting problem of distilling knowledge from video world models to enhance downstream perception tasks, such as geometric and semantic understanding.
* The pertaining scheme operates on both images and outputs from foundation models (e.g., depth anything output, edge map) to capture rich visual information across multiple modalities.
* Extensive experiments are conducted to examine the scaling property of the proposed model, such as model capacity and temporal context length in autoregressive next-token prediction during pretraining.
* Results show that the pre-trained model improves downstream tasks, including LiDAR depth prediction and image semantic segmentation, across multiple datasets.

**Weaknesses:**

* The downstream tasks primarily involve low-level vision tasks, similar to the pre-training inputs. For example, LiDAR depth prediction in the downstream task is akin to the depth map input. However, how does the model perform on high-level vision tasks, such as detection and prediction? An interesting aspect of world models is that if they can predict the future, they may have learned the behaviour of actors in the scene. Distilling this kind of intelligence could be particularly valuable.
* This paper also lacks comparisons with other baselines. The authors should benchmark their method against state-of-the-art depth prediction and semantic segmentation methods on the dataset to assess the performance gap.

**Questions:**

See weakness above.

---

### Meta-Review · Area_Chair_d3ZL · 2024-12-22

**Metareview:**

This paper introduces a promising method, PerceptionLM, leveraging pre-trained world models for low-level perception tasks. This paper formulates multi-modal next token prediction as a world model learning problem and argues that learning from this task can greatly improve the spatial understanding for downstream tasks. This paper provides experiments on low-level perception tasks.

After reading reviewers' comments, there is a clear consensus that this paper 's focus on low-level perception is not convincing nor representative of all perception tasks. AC agrees with reviewers that a semantic understanding task (such as object detection) is even more useful and practical in an autonomous driving pipeline. Also, this paper claims a general world model learning framework, which is only tested on autonomous driving scenarios. Authors should provide more general test cases if sticking to this general claim. Otherwise, they should consider reducing the scope to autonomous driving.

**Additional Comments On Reviewer Discussion:**

Reviewers raised several detailed questions regarding some design choices, which are addressed by the additional experiments provided in the rebuttal process. However, authors fail to provide object detection results, which are extremely important and common in autonomous driving. Therefore, AC believes that the rebuttal fails to address all reviewer concerns.

---

### Decision · Program_Chairs · 2025-01-22

Reject